# Temperature Scaling in Discrete Sequence (Language) Models

**Hannah Scheufele** [* † 1]  **Peter Blohm** [* † 2]  **Vikas Garg** [3 4]

## Abstract

Temperature scaling is widely used to control generation behavior and improve reasoning performance in discrete sequence models. However, temperature scaling at the sequence level is particularly challenging since the partition function of the model is typically not tractable and naive *token-wise* temperature scaling induces bias in the maximum a posteriori (MAP) estimates. Discrete diffusion settings exacerbate this issue: enhanced flexibility due to multiple potential orderings for generating tokens complicates likelihood computation and incurs high variance.

We propose and investigate fine-tuning objectives for sequence-level temperature scaling, along with metrics to quantify the success of a given temperature scaling procedure. Specifically, our contributions are threefold. First, we introduce our objective across a wide class of sequence models through a novel formalism that unifies autoregressive and diffusion-based language models. Second, we design two metrics that quantify temperature scaling based on likelihood *ratios* and thus obviate access to the partition function. Finally, we introduce fine-tuning objectives that reliably achieve the desired change in model temperature.

Our experiments with language models (GPT2, BD3LM) show that the proposed approach leads to more consistent generation with lower perplexity. Furthermore, we provide empirical evidence that it can enhance the reasoning performance in language models. Code is available at [github.com/Aalto-QuML/temp_lm](github.com/Aalto-QuML/temp_lm).

---

[*]Equal contribution [†]Work done during an internship at Aalto University. [1]Georg-August-Universität Göttingen, Germany [2]TU Wien, Austria [3]Aalto University, Finland [4]YaiYai Ltd. Correspondence to: Hannah Scheufele <hannah.scheufele@aalto.fi>, Peter Blohm <peter.blohm@tuwien.ac.at>, Vikas Garg <vgarg@csail.mit.edu>.

*Proceedings of the 43$^{rd}$ International Conference on Machine Learning*, Seoul, South Korea. PMLR 306, 2026. Copyright 2026 by the author(s).

| | |
|---|---|
| **Unifying Formulation for Sequence Models** | Section 2 |
| **Likelihood Calculation for Sequence Models** | Section 4 |
| **Partition-Free Temperature Scaling Metric** | Section 5.1 |
| **Sequence Temperature Scaling Objectives** | Section 5.2 |
|    Importance Sampling Temperature Scaling | 5.2.1 |
|    Temperature Variance Regularization | 5.2.2 |
|    Scalable Variance Regularization | 5.2.3 |
| **Experiments** | Section 6 |

*Table 1.* Overview of the main contributions in this work.

## 1. Introduction

Generative sequence models are widely used in modern machine learning. They solve generation problems by factorizing over subproblems, i.e., tokens, allowing them to tackle complex tasks such as language processing. Different models perform this factorization in different ways: autoregressive models, the current state-of-the-art method, do so sequentially (Radford et al., 2019; Brown et al., 2020; Vaswani et al., 2023), while discrete diffusion models (Austin et al., 2023; Campbell et al., 2022; Lou et al., 2024; Sahoo et al., 2024; Shi et al., 2024; Ou et al., 2025; Nie et al., 2025; Ye et al., 2025) operate in a random order. Hybrid methods that combine some aspects of autoregressive and discrete diffusion have also been proposed (Sahoo et al., 2025; Arriola et al., 2025; Wang et al., 2025; Wu et al., 2025b;a; Song et al., 2025).

A standard approach to modulate the behavior of any model, including sequence models, is *temperature scaling* (TS). The idea of temperature scaling is to adjust the sharpness of the model distribution, e.g., to improve performance in downstream tasks or control generation diversity. More formally, one can anneal the base model distribution $p$ by a scalar $T \in \mathbb{R}_{>0}$, so that the resulting model $q_T \propto p^{1/T}$. For $T < 1$, the distribution is sharper, pushing probability mass toward higher-likelihood outputs; conversely, for $T > 1$, the distribution is flatter, yielding a higher diversity of outputs.

While temperature scaling is intuitive and commonly leveraged, subtle caveats are often overlooked in its motivation and application. This work is motivated by a fundamental question: *why may we want to use temperature scaling in*

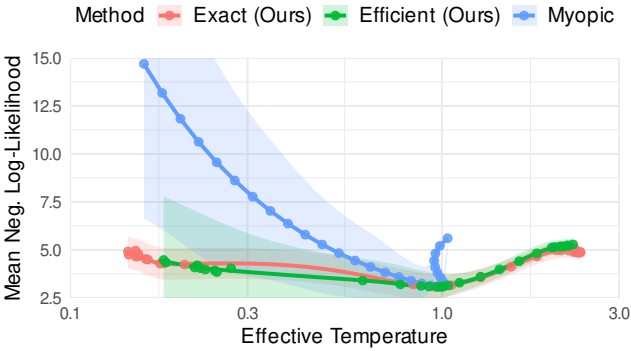

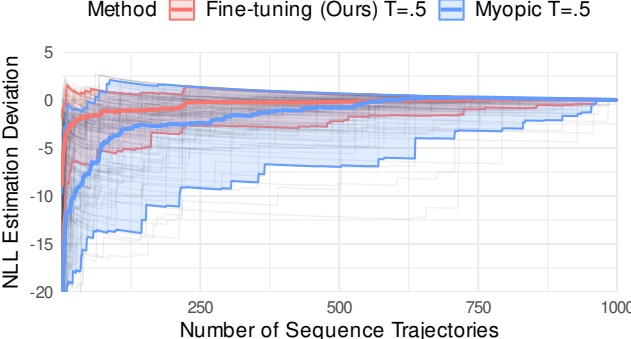

*(a)* Comparison between our proposed methods and token-wise (myopic) temperature scaling. Models were biased towards different temperatures. We show mean NLL (lower is better) and variance-indicating ribbons. Fine-tuning is able to scale to low *and* high temperatures with reduced performance loss and lower variance.

*(b)* Running estimated NLL for 64 different sequences with an increasing amount of unmasking trajectories. NLL estimates are highly variable, even with many sampled trajectories. Fine-tuning shows drastically lower variance than myopic temperature scaling.

*Figure 1.* Comparison of our procedure versus token-wise temperature scaling with BD3LM (Arriola et al., 2025), measured on the Openwebtext dataset (Gokaslan & Cohen, 2019).

*generative modeling?* For some ambient space $\mathcal{X}$, most generative tasks have a notion of *good* elements that we would like to sample from. The common assumption is that model likelihoods correlate well with sample quality, so if we can increase the average model likelihood, we presumably increase the average quality as well. For example, when we assume there exists a set of uniformly *good*, high-density elements $A \subset \mathcal{X}$ in the space $\mathcal{X}$ of all possible outputs, the goal of temperature scaling perfectly aligns with the question *"how can we increase $p(A)$?"* For any temperature-scaled distribution $p_T = p^{1/T}(x)/Z_T$ with $0 < T \leq 1$ and $Z_T = \int p^{1/T}$ as the partition function, we have that

$$p(x) > p(x') \iff \frac{p_T(x)}{p_T(x')} > \frac{p(x)}{p(x')}. \tag{1}$$

For the high-density set $A$, we then have $p_T(A) > p(A)$.

Inconveniently, approximating $p_T$ from $p$ is non-trivial in *sequence models* specifically, as we need to find $q(x) \propto p^{1/T}(x)/Z_T$, where $Z_T = \int p^{1/T}$ is the partition function. Karan & Du (2026) show that sharpening the distribution $p$ using their (expensive) sequence-level temperature scaling procedure is effective in improving its reasoning capabilities.

Traditionally, temperature scaling is instead done at the token level, scaling the probability of individual tokens without considering the sequence as a whole. This simple approach can lead to miscalibrated sequence distributions, causing problems such as overconfidence or sequence degeneration (Guo et al., 2017; Holtzman et al., 2019). This instability is illustrated in the blue curve of Figure 1a, which shows a decrease in performance for $T \neq 1$. While there exist multiple methods that aim to control sequence likelihoods (Joy et al., 2023; Ding et al., 2020; Shen et al.,

2024; Choi et al., 2024), they do not offer any theoretical interpretation for temperature scaling (Shih et al., 2023; Karan & Du, 2026).

In this work, we investigate sequence-level temperature scaling, meaning scaling $p(x)$ directly. To achieve this, we introduce a new metric for measuring a model's sequence-level temperature. We also present two new TS fine-tuning procedures. Henceforth, unless specified otherwise, we use TS to refer to sequence-level temperature scaling.

When we try to learn a model $q \propto p_T$, the right objective is non-trivial. First, $p_T$ is generally only accessible up to the unknown partition function $Z_{p_T}$, which restricts which techniques can be used. Second, to scale the likelihood of a sequence, the likelihood must first be well-defined. However, in most sequence-generation models, the likelihood of a sequence is not uniquely determined by the model parameters—it depends on the parameters of the sampling process itself. In most state-of-the-art discrete diffusion models (Sahoo et al., 2024; Shi et al., 2024; Ou et al., 2025; Nie et al., 2025), the order of generation itself is subject to randomness, so the same sequence can be sampled in many different ways. This leads to three issues specific to stochastic generation ordering.

- Models exhibit very high variance in their estimated likelihoods for a given sequence, depending on the trajectory of the generation process, meaning that simply fixing this order biases the model's estimates. This effect is illustrated in Figure 1b, which shows the drastic variability of order-dependent likelihood estimates.

- Discrete diffusion language models do not restrict generation orders, which leads to a combinatorial explosion for computing expected likelihood over all

trajectories. For this reason, discrete diffusion models commonly only use an upper bound on the likelihood.

- Sequence models follow a variety of different formalisms. Some rely on discrete or continuous time and define noising processes (Austin et al., 2023; Campbell et al., 2022) while others use order-dependent formalisms (Radford et al., 2019), and parallel generation of multiple tokens may or may not be allowed.

To tackle these issues, we formalize exact likelihood computation across a wide array of generative sequence models by recasting them as a unified class. With this result, we then devise theoretically well-founded partition-free metrics to quantify effective TS at the sequence level. In an analog to Long Horizon Temperature Scaling (LHTS) (Shih et al., 2023), we present a KL-minimizing loss function, built on importance sampling. To mitigate the high variance of importance sampling, we propose additional regularization schemes that greatly improve training stability *even without relying on importance sampling*. Our objectives are efficient and well-behaved, allowing us to use intermediate model checkpoints during training as interpolations between the baseline and target temperatures.

We evaluate our method on block diffusion (Arriola et al., 2025) and GPT2 (Radford et al., 2019). We show that this scheme (i) achieves effective temperature scaling at the sequence level and (ii) has a better performance than token TS. Furthermore, we show that our method can lead to better results on reasoning benchmarks for Phi-3.5-mini-instruct (Abdin et al., 2024) and Qwen2.5-7B (Qwen et al., 2025). An overview of the structure of this paper and our contributions can be seen in Table 1.

## 2. Unified Notation for Language Models

Notation for language models varies across literature and assigns different semantics to the "order of generation". In this work, we present a general and lightweight formalism that unifies generative sequence models. We denote sequences of length $L$ over a given vocabulary $\mathcal{X}$ with $x \in \mathcal{X}^L$, and for an index $i \leq L$ denote the $i$-th token with $x_i \in \mathcal{X}$, for a given index-set $I \subset \{1, \ldots, L\}$ denote the subsequence $(x_i \mid i \in I)$ with $x_I \in \mathcal{X}^{|I|}$, and the *prefix* $(x_1, \ldots, x_{i-1})$ with $x_{<i} \in \mathcal{X}^{i-1}$. For integers $n_1, n_2 \in \mathbb{N}, n_1 \leq n_2$, we further denote with $[n_1] = \{1, \ldots n_1\}$ and with $[n_1, n_2] = \{n_1, \ldots n_2\}$. A generative sequence model generates sequences according to a (possibly stochastic) *unmasking schedule*.

**Definition 2.1** (Unmasking Schedule). Let $x \in \mathcal{X}^L$ be a sequence over the vocabulary $\mathcal{X}$ of length $L$. We then define a $K$-step unmasking schedule $\pi$ as

$$\pi : [0, K] \to \mathcal{P}([L])$$

where $\pi_0 = \emptyset$ and $\pi_K = [L]$. We restrict $\pi$ to be strictly monotone, i.e., $\forall s, t \in [1, K]$ with $s < t$ it holds that $\pi_s \subsetneq \pi_t$. $\pi_t$ is then the set of positions whose token values are fixed after step $t$.

We then use $x_{\pi_t}$ for the partially unmasked sequence containing the tokens at positions in $\pi_t$, and denote the token indices unmasked at step $t$ with $\Delta_t(\pi) := \pi_t \setminus \pi_{t-1}$. For the rest of the paper, we use the shorthand $\Delta_t = \Delta_t(\pi)$, unless ambiguous. For a given model $p$, we denote the probability of a sequence $x$ being generated by $p$ under $\pi$ as

$$p(x; \pi) := \prod_{t=1}^{K} p(x_{\Delta_t} \mid x_{\pi_{t-1}}) \tag{2}$$

similar to Kim et al. (2025).

When the unmasking schedule is stochastic, we denote it by the random variable $\Pi$. With this definition, we can characterize the following model classes solely in terms of different distributions over $\Pi$, i.e., by their different strategies for unmasking tokens and, consequently, by different possible factorizations of $p(x)$.

## 3. Background

With our canonical notation, we can now define typical model types for sequence generation in a standardized, interoperable way.

### 3.1. Autoregressive Models

Autoregressive models are the state of the art for sequence generation, particularly in language generation (Radford et al., 2019; Brown et al., 2020; Vaswani et al., 2023; Zheng et al., 2024). For any auto-regressive model $p$ the probability of a given sequence $x \in \mathcal{X}^L$ factorizes as a product of conditionals over prefixes. To express index-based autoregression as a schedule, we define $\pi^{\text{AR}}$ with $\pi_t^{\text{AR}} = [t]$ and $\Delta_t = \{t\}$.

$$p(x; \pi^{\text{AR}}) = \prod_{t=1}^{L} p(x_{\Delta_t} \mid x_{\pi_{t-1}^{\text{AR}}}) = \prod_{i=1}^{L} p(x_i \mid x_{<i}). \tag{3}$$

### 3.2. Masked Diffusion

Masked diffusion models (MDMs), as described in most literature, define a forward noising process over discrete or continuous time that stochastically masks token positions (Austin et al., 2023; Campbell et al., 2022; Lou et al., 2024; Sahoo et al., 2024; Shi et al., 2024; Ou et al., 2025). The MDM is then trained to reverse this time-dependent process. Time dependence in MDMs is realized *exclusively* via the noising schedule $\alpha : [0, 1] \to [0, 1]$. The schedule $\alpha$ defines the unmasking probability for a masked token at time $\tau$, i.i.d. across indices. For this paper, we define an MDM to

be any model where $\Pi^\alpha$ follows an unrestricted distribution according to some i.i.d. schedule $\alpha$.

MDMs are trained using a negative ELBO to maximise the expected log-likelihood of sequences in a dataset, uniformly over unmasking *permutations* (Uria et al., 2014; Ou et al., 2025; Kim et al., 2025)

$$\mathcal{L}_{\text{Diff}} = -\mathbb{E}_{x \sim \mathcal{D}, \Pi \sim \mathcal{U}(S_L)} \left[ \sum_{t=1}^{K} \log p(x_{\Delta_t} \mid x_{\Pi_{t-1}}) \right], \quad (4)$$

where $\mathcal{U}(S_L)$ denotes a uniform distribution over permutations of length $L$. At sampling time, MDMs might unmask multiple tokens at the same time $t$, where the conditional independence of all tokens in $\Delta_t$ is assumed, i.e.,

$$p(x_{\Delta_t} \mid x_{\pi_{t-1}}) = \prod_{i \in \Delta_t} p(x_i \mid x_{\pi_{t-1}}). \quad (5)$$

This *parallel unmasking* improves sampling efficiency by reducing the effective number of sampling steps. However, the independence assumption introduces a distribution shift and might drastically decrease model performance (Kang et al., 2025; Chang et al., 2022; Austin et al., 2023).

### 3.3. Block Diffusion

Block diffusion models (Arriola et al., 2025; Wu et al., 2025b;a; Wang et al., 2025) are a restricted class of MDMs in which sequences are partitioned into (fixed-length) *blocks* of length $B \ll L$. In block diffusion, any unmasking schedule *within* a given block is allowed, but different blocks are generated *in sequence*. More formally, the stochastic unmasking schedule $\Pi^B$ in block diffusion only has support on *block-respecting* $\pi$. We call $\pi$ block-respecting if it can be defined as a concatenation of block schedules $\pi^{B_i}$ such that

$$p(x; \pi) = \prod_i p(x_{B_i} \mid x_{B_{<i}}; \pi^{B_i}). \quad (6)$$

This hybrid approach increases flexibility in sequence length and enables drastically more efficient training.

### 3.4. Temperature Scaling

A central technique for controlling the trade-off between diversity and quality in sequence models is *temperature scaling* (Guo et al., 2017; Holtzman et al., 2019; Desai & Durrett, 2020; Li et al., 2025).

Given a density $p$ over some space $\mathcal{X}$ we want to approximate for some fixed temperature $T \in \mathbb{R}_+$ the annealed distribution $p_T$ where

$$p_T(x) = p(x)^{1/T} / Z_{p_T}. \quad (7)$$

Here $Z_{p_T} = \int p(x)^{1/T}$ is the normalizing partition function of $p_T$. Renormalization is required for temperature

scaling, but for many (non-sequence) model classes, this can be done implicitly via application of the softmax function, allowing for trivial one-parameter temperature scaling (Guo et al., 2017).

**Naive Temperature Scaling on Sequence Models** When the domain $\mathcal{X}$ is small, computing $Z_{P_T}$ is non-problematic. However, for language models, the vocabulary *of each token* already contains thousands of logits. Computing $Z_{p_T}$ explicitly for sequences of length $L$ requires normalization over $|\mathcal{X}|^L$ possible outputs and is completely intractable.

Shih et al. (2023) show that token-wise, or *myopic* temperature scaling fails on sequences. Denoting for some sequence $x$ with $Z_{p_T}^{\Delta_i} = \int p(x_{\Delta_i} \mid x_{\pi_{i-1}})$ the tractable computable token-wise partition functions, it holds that

$$p_T(x; \pi) \propto \frac{\prod_i p(x_{\Delta_i} \mid x_{\pi_{i-1}})^{1/T}}{Z_{p_T}}$$

$$\not\propto \prod_i \frac{p(x_{\Delta_i} \mid x_{\pi_{i-1}})^{1/T}}{Z_{p_T}^{\Delta_i}}.$$

This mismatch makes the naive approach to temperature scaling inapplicable for sequence models, and introduces known pathologies Holtzman et al. (2019).

**Long Horizon Temperature Scaling** Shih et al. (2023) tackle sequence-level temperature scaling for autoregressive models with the fine-tuning procedure *long horizon temperature scaling* (LHTS). LHTS trains the temperature-scaled sequence model $q_T$ to approximate the annealed fixed model $p_T$ with likelihood maximization under importance sampling. Formally, LHTS minimizes

$$\mathbb{E}_{x \sim p_T}[-\log q_T(x_i \mid x_{<i})]$$
$$= \mathbb{E}_{x \sim p}[-w_T(x_i) \log q_T(x_i \mid x_{<i})], \quad (8)$$
$$w_T(x_i) = \exp\left( \frac{1-T}{T} \log p(x \mid x_{<i}) - \log Z_{p_T} \right).$$

Here $Z_{p_T}$ is a constant and does not need to be computed. This objective can be shown to minimize the KL-divergence $\text{KL}(p_T \| q_T)$. However, for sequence modeling, the formulation is limited to autoregressive models. Furthermore, the high variance of the importance weights requires a careful training regimen to manage stability at the cost of biasing the objective (Shih et al., 2023).

## 4. Stochastic Sequence Unmasking Models

To apply TS to a given model, we need a well-defined notion of the likelihood of a sequence. This is nontrivial in stochastic settings, where the same sequence can be generated by different schedules, and the probability across schedules can vary widely, as shown in Figure 1b.

In this section, we define the class of stochastic sequence unmasking models (SSUMs). This general class is compatible with a wide body of the existing literature on auto-regressive *as well as* discrete diffusion models. We then present an approach based on dynamic programming to compute the likelihood of the sequence under a given model, conditioned on its scheduling distribution.

### 4.1. Stochastic Sequence Unmasking Models

Different sequence models are usually viewed through distinct formalisms. We provide a unified perspective for many models, viewing progressive unmasking of tokens as their common task (albeit according to different schedules). Notably, in contrast to another unified formalism by Fathi et al. (2025), our formulation is time-independent and includes some time-dependent MDMs.

**Definition 4.1** (Stochastic Sequence Unmasking Model). Let $p(x_i \mid x_S)$ be a sequence model that predicts a token $x_i$ at position $i \notin S$, conditioned on an observed partial sequence $x_S$. Let $\Pi$ be a random variable over unmasking schedules. We say $p$ is a *stochastic sequence unmasking model* (SSUM) if for all $\pi$ with $\Pr(\Pi = \pi) > 0$:

- $\pi$ is strictly monotone, i.e., each position is unmasked exactly once (no remasking).

- $p(x_i \mid x_S)$ depends only on $i$ and $x_S$.

- $\Pi$ is independent of the token values $x$.

Autoregressive models are SSUMs as $\Pr(\Pi = \pi^{\mathrm{AR}}) = 1$ and therefore $\Pr(\Pi \neq \pi^{\mathrm{AR}}) = 0$ and $\pi^{\mathrm{AR}}$ satisfies the SSUM conditions. Furthermore many MDMs are SSUMs as the following proposition shows.

**Proposition 4.2.** *Absorbing D3PM (Austin et al., 2023), MDLM (Sahoo et al., 2024), MD4 (Shi et al., 2024), RADD (Ou et al., 2025) are SSUMs.*

Calculating the probability under a model $p$ can be difficult, as the probability using different unmasking schedules $\pi, \pi'$ might not be the same $p(x; \pi) \neq p(x; \pi')$ and for longer sequences, the variance between different schedules can vary significantly (Uria et al., 2014; Ou et al., 2025). Instead, the expectation over the unmasking schedules can be used to compute the probabilities.

*Remark* 4.3. Let $p$ be a SSUM, then the probability of a sequence $x \in \mathcal{X}^L$ can be calculated by marginalizing over the unmasking schedules $\pi$

$$p(x) = \mathbb{E}_\Pi[p(x; \Pi)]. \qquad (9)$$

**Corollary 4.4.** *Absorbing D3PM, MDLM , MD4, RADD over* continuous time *unmask according to $\Pi \sim \mathcal{U}(S_L)$, i.e.,*

*a sequence $x \in \mathcal{X}^L$ is unmasked sequentially and*

$$\Pr(x) = \mathbb{E}_{\Pi \sim \mathcal{U}(S_L)}[p(x; \Pi)] = \frac{1}{L!} \sum_\pi p(x; \pi). \qquad (10)$$

As autoregressive models and MDLMs are SUMMs, it follows that Block diffusion models are also SSUMs, as they are built on these models and changing only the distribution of $\Pi$, which is in continuous time $\mathcal{U}(S_B)^M$, i.e., $\Pr(\pi) = \frac{1}{B!^M}$.

While we can compute the marginal likelihood, it unfortunately does not admit unbiased approximation in log-space:

*Remark* 4.5 (No unbiased estimator for the log marginal likelihood). For fixed $x$, and arbitrary $\Pi$ and $p$ satisfying $0 < p(x; \pi) < 1$, there is no universal unbiased finite-sample estimator of

$$\log p(x) = \log \mathbb{E}_\Pi\big[p(x; \Pi)\big].$$

That is, one cannot in general estimate the log of an expectation without bias from finite samples $\pi_1, \ldots, \pi_n \sim \Pi$.

In the following section, we discuss methods to compute $\log p(x)$ as efficiently as possible, and later utilize a biased Monte-Carlo estimator for $\log p(x)$.

### 4.2. Exact Computation of Sequence Likelihoods

For our fine-tuning objective, we now need to be able to compute $p(x)$ for a given SSUM $p$ and a given sequence $x \in \mathcal{X}^L$. Unfortunately, in the general case, the support of $\Pi$ is exponential in $L$ for MDMs. Allowing for parallel unmasking, the number of possible unmasking schedules is equal to the number of different *weak orders* on $L$ elements, also known as the *ordered Bell numbers*, which are in $\mathcal{O}(L!1.5^L)$. We can, however, drastically improve upon this pessimistic bound in *block diffusion* with *continuous time*.

**Proposition 4.6.** *Let $p$ be a block diffusion model, with block size $B$, in a sequential unmasking regimen. Then, for a given $x$, we can compute $p(x)$ in time $O(L2^B)$.*

This faster computation can be achieved by using a dynamic program that marginalizes over partial sequences rather than over unmasking sequences. As a notational simplification, we define an *extension* of a schedule at time $t$ as

$$E(\pi, t) = \{\pi' \in \mathcal{P}([L])^L \mid \forall t' \leq t : \pi'_{t'} = \pi_{t'}\}. \qquad (11)$$

For block diffusion we can calculate the probability of each partial block $x_{\pi_t^{B_i}}$ using the probabilities $p(x \mid x_{\pi_{t+1}^{B_i}})$ for $\pi_{i+1}^{B_i} \in E(\pi^{B_i}, i)$ in the following way

$$p(x \mid x_{\pi_i}) = \frac{1}{L - i} \sum_{\pi_{i+1} \in E(\pi, i)} p(x \mid x_{\pi_{i+1}}) p(x_{\Delta_{i+1}} \mid x_{\pi_i})$$

$$(12)$$

| Model | Sampling Method | Distributions | Models |
|---|---|---|---|
| Parallel unmasking MDM | any order, parallel | $\Pi^{\alpha}$ | Absorbing D3PM (Austin et al., 2023), MDLM (Sahoo et al., 2024), MD4 (Shi et al., 2024), RADD (Ou et al., 2025) |
| Sequential unmasking MDM | any order, sequential | $\mathcal{U}(S_L)$ | same as above |
| Autoregressive Model | In order, sequential | $\delta_{\pi^{AR}}$ | GPT (Radford et al., 2019) |
| Block diffusion | block-respecting order | $\Pi^{B_i,\alpha}, \mathcal{U}(S_B^M)$ | BD3LM (Arriola et al., 2025) |

*Table 2.* Overview of models, sampling methods, distributions, and references.

starting with $p(x|x_{\pi_L}) = 1$. This computes the probability of each partial $x_{\pi_t^{B_i}}$ once resulting in a runtime of $\mathcal{O}(L2^B)$. Algorithm 1 gives the dynamic program for block diffusion.

---

**Algorithm 1** Dynamic program for computing $p(x)$ exactly

---

1: **Input:** Block diffusion model $p$, sequence $x \in \mathcal{X}^L$
2: **Output:** $p(x), p(x \mid x_{\pi_i})$
3: $p(x \mid x_{\pi_L}) = 1$
4: **for** each block $B_i$ **do**
5:    **for** $t = B - 1, \ldots, 0$ **do**
6:       **for** each $\pi_t = \pi_t^{B_i}$ **do**
7:          $p(x \mid x_{\pi_i}) = \ldots$ {Equation (12)}
8:       **end for**
9:    **end for**
10:    $p(x_{B_i} \mid x_{<B_i}) = p(x_{B_i} \mid x_{\pi_0})$
11: **end for**
12: $p(x) = \prod_i p(x_{B_i} \mid x_{<B_i})$

---

## 5. Stochastic Sequence Temperature Scaling

In this section, we introduce Stochastic Sequence Temperature Scaling (SSTS) as a technique for fine-tuning a model's sequence-level temperature. As an initial contribution, we introduce metrics to validate a model's sequence-level temperature (SST) without access to its partition function. Building on these fundamental concepts, we introduce three distinct objective functions for SSTS.

### 5.1. Partition-Free Temperature Scaling Metrics

As we established in the Section 1, temperature-scaling does not change the likelihood ranking of $p$:

$$p(x) > p(x') \iff q_T(x) > q_T(x')$$

To measure the agreement between the rankings induced by $q_T$ and $p$ on some dataset of sequences $\mathcal{D}$, we use Kendall's $\tau$, a standard measure of rank (Kendall, 1938),

$$\tau = \frac{1}{|\mathcal{D}|^2} \sum_{x,x' \in \mathcal{D}} \text{sgn}\left(\log \frac{p(x)}{p(x')}\right) \text{sgn}\left(\log \frac{q_T(x)}{q_T(x')}\right). \tag{13}$$

Because the logarithm is monotonic, the rankings are preserved in log space. Moreover, taking log-ratios cancels the partition functions $Z_p$ and $Z_{q_T}$, so unnormalized models are sufficient. The rankings induced by $p$ and $q_T$ perfectly agree when $\tau = 1$, and completely disagree when $\tau = -1$.

Rank-preservation alone is an incomplete metric for temperature scaling. Inconveniently, the precise numeric relation between $p$ and $q_T$ is influenced by the unknown partition function $Z_{p_T}$. This unknown, however cancels when we investigate likelihood *ratios*

$$q_T \propto p \implies \frac{q_T(x)}{q_T(x')} = \left(\frac{p(x)}{p(x')}\right)^{1/T} = \frac{\frac{p^{1/T}(x)}{Z_{p_T}}}{\frac{p^{1/T}(x')}{Z_{p_T}}}. \tag{14}$$

We utilize this observation and refine our objective as

**Lemma 5.1** (Temperature Scaling via Likelihood Ratios). *Let $p, q_T$ be probability densities over some space $\mathcal{X}$. It holds that $q_T \propto p^{1/T}$ iff*

$$\forall x, x' \in \mathcal{X}^L : \log \frac{p(x)}{p(x')} = T \log \frac{q_T(x)}{q_T(x')}. \tag{15}$$

For two models $p, q_T$, define their *temperature log-ratio* for a given sequence $x$ as

$$h(x) \coloneqq \log p(x) - T \log q_T(x) \tag{16}$$

and try to fine-tune a model $q_T$ using our procedure such that $\forall x : h(x) \approx 0$, while keeping the variance of the ratio across sequences low. To estimate the resulting temperature change, we can use linear regression on the log-likelihoods.

**Definition 5.2** (Effective Temperature). Let $p, q_T$ be probability densities over some space $\mathcal{X}$, then we estimate the *effective temperature* of $q_T$ on a dataset $\mathcal{D}$ with homogeneous least squares regression:

$$T_{\text{eff}} \coloneqq \frac{\sum_{x,x' \in \mathcal{D}} \log \frac{p(x)}{p(x')} \log \frac{q_T(x)}{q_T(x')}}{\sum_{x,x' \in \mathcal{D}} \left(\log \frac{q_T(x)}{q_T(x')}\right)^2} \tag{17}$$

## 5.2. Fine-Tuning Objectives for SSTS

After defining temperature ratios as a metric for the effectiveness of TS, we now extend sequence-level temperature-scaling loss functions to any SSUM, i.e., any model that adheres to Definition 4.1. During this section, we define $p$ to be a fixed pre-trained SSUM and compute $p(x)$ according to Section 4.2 as well as all relevant conditional probabilities over sub-sequences. We investigate the following three distinct temperature-scaling objectives with complementary characteristics.

### 5.2.1. IMPORTANCE SAMPLING SSTS

To extend Vanilla LHTS in Equation (8) to the stochastic un-masking schedules of masked diffusion models, we augment the diffusion objective from Equation (4) with importance sampling. We can define our SSTS loss as

$$\mathcal{L}_{\text{SSTS}} = - \mathop{\mathbb{E}}_{\pi \sim \mathcal{U}(S_L)} \left[ \sum_{t=1}^{K} w(x_{\pi_t}) \log q_T(x_{\Delta_t} \mid x_{\pi_{t-1}}) \right] \tag{18}$$

with weights and baselines defined as

$$w_T(x_{\pi_t}) = \exp \left( \frac{1-T}{T} \log p(x \mid E(\pi, i)) - \mu(x_{\pi_t}) \right)$$

$$\mu(x_{\pi_i}) = \frac{1}{|\mathcal{D}|} \sum_{x \in \mathcal{D}} \frac{1-T}{T} \log p(x \mid E(\pi, i)).$$

The conditional probability under the base model can be calculated with the unmasking schedule, as we show in Section 4.2 using $p(x) = \mathbb{E}_\Pi \left[ p(x; \Pi) \right]$ or for conditional probabilities $p(x \mid x_{\pi^*(i)}) = \mathbb{E}_\Pi[p(x; \Pi \mid x_{\pi^*(i)})]$ where we only use unmasking schedules $\pi$ that go through $\pi^*(i)$.

Under mild conditions, we can show that $\mathcal{L}_{\text{SSTS}}$ is a well-behaved loss function for our objective.

**Proposition 5.3.** *Let $p$ be a pre-trained MDM, sampled according to $\Pi$, and $\mathcal{D} \subset \mathcal{X}^L$ be a data set of sequences. If $\forall x \in \mathcal{D}$ and $\forall \pi_i$ with $\Pr_\Pi(\pi_i) > 0$ it holds that $\mu(x_{\pi_i})$ is finite, then $\mathcal{L}_{\text{SSTS}}$ minimizes the KL-Divergence $KL(p_T \| q_T)$ and is a strictly proper loss, it is minimal only if $q_T \propto p^{1/T}$.*

This importance-sampling-based approach allows us to define a loss function that minimizes the KL-divergence (Cover & Thomas, 2006) to our target distribution; the exponential weights $w$ exhibit high variance in practice, due to the inherently high variance of $\log p(x \mid x_{\pi_i})$. In their paper, Shih et al. (2023) utilize a variety of variance-reducing techniques to stabilize training. Among these techniques, they resort to clipping high weights. This, or similar, approaches are hard to avoid when using importance sampling, but they introduce bias into the objective (Elvira et al., 2019; Korba & Portier, 2021). To mitigate these weaknesses, we propose a regularization scheme that minimizes $h$'s variance and drastically stabilizes training.

### 5.2.2. ENTROPY VARIANCE REGULARIZATION

The loss function $\mathcal{L}_{\text{SSTS}}$ we introduced in Section 5.2.1 is strictly proper, but unstable in practice. $\mathcal{L}_{\text{SSTS}}$ can be equivalently expressed as minimizing the expected temperature log-ratio $\mathbb{E}_{p_T}[h]$. To stabilize training, we introduce a regularization scheme that encourages $h$ to be constant, i.e., minimizing $\text{Var}_{p_T}[h]$. We do this with regression towards the equality in Equation (15), and define our *ratio regularizer* as

$$\mathcal{L}_{\text{ratio}} = \mathbb{E}_{p_T} \left[ \left( \log \frac{p(x)}{q_T(x)^T} - \log \frac{p(x')}{q_T(x')^T} \right)^2 \right] \tag{19}$$

$$= \mathbb{E}_{x,x' \sim p_T} \left[ (h(x) - h(x'))^2 \right] \tag{20}$$

This contrastive objective resembles the least squares formulation of concrete score matching (Meng et al., 2023) or ratio matching (Hyvärinen, 2007). In fact, Hyvärinen (2007) define ratio matching with a transform $g(u) = \frac{1}{1+u}$ (with adaption to our setting) as

$$\mathbb{E}_{p_T} \left[ \left( g \left( \frac{p(x)}{p(x')} \right) - g \left( \frac{q_T(x)^T}{q_T(x')^T} \right) \right)^2 \right], \tag{21}$$

which minimizes the Brier-Score between $p_T$ and $q_T$. Remarkably, this is equivalent to applying the sigmoid function $\sigma(u) = \frac{1}{1+\exp(-u)}$ to the log-probability ratios in our objective $\mathcal{L}_{\text{ratio}}$. Both objectives minimize a Bregman divergence between likelihood ratios, but exhibit slightly different characteristics in optimization. We now formalize the characteristic of $\mathcal{L}_{\text{ratio}}$.

**Proposition 5.4.** *Let $p$ be a pretrained SUMM with global support over $\mathcal{X}^L$, and let $T \in \mathbb{R}_+$ be the target temperature. A model $q_T$ that minimizes $\mathcal{L}_{\text{ratio}}$ minimizes the* rel-*ative entropy variance $V(p_T \| q_T)$ (Hayashi, 2009). When $V(p_T \| q_T) = 0$, it holds that $KL(p_T \| q_T) = 0$.*

Unfortunately, Proposition 5.4 does not guarantee small KL divergence for *every* minimizer of $\mathcal{L}_{\text{ratio}}$ without additional assumptions. Intuitively, the main issue is *likelihood drift*, where $q_T$ might assign arbitrarily small likelihoods to sequences with high probability under $p$. We drift by using $\mathcal{L}_{\text{ratio}}$ only as a *regularizer*, combined with the default (tokenwise) likelihood maximization objective in Equation (4).

The next issue to address in $\mathcal{L}_{\text{ratio}}$ is, as in $\mathcal{L}_{\text{SSTS}}$, importance sampling. Equation (20) is defined over $p_T$, which we cannot directly sample from.

As a change of measure from $p_T$ to $p$, we would need to reweight each element in this expectation by $w(x)w(x')$, where $w$ are the importance weights, i.e.,

$$w(x) = \exp \left( \frac{1-T}{T} \log p(x) - \log Z_{p_T} \right). \tag{22}$$

While we cannot provide simple, meaningful bounds for $w(x)$ nor the product, we assume that in practice, our actual

fine-tuning dataset $\mathcal{D}$ contains sequences with high likelihood, i.e., $w(x) = p_T(x)/p_{\mathcal{D}}(x)$ is bounded. Under this empirically measurable assumption, our regularizer $\mathcal{L}_p = \text{Var}_p[h]$ remains a good approximation for $\mathcal{L}_{\text{ratio}}$. Importantly, when $p$ has global support $\mathcal{L}_p = 0$ implies $\mathcal{L}_{\text{ratio}} = 0$.

### 5.2.3. MONTE-CARLO REGULARIZATION

In the previous section, we motivate that our regularization scheme with $\mathcal{L}_{\text{ratio}}$ can perform reasonably well, even when importance weights are omitted. $\mathcal{L}_{\text{ratio}}$ pushes the exact probability $q_T(x) = \mathbb{E}_\Pi[q_T(x; \pi)]$ close to $p^{1/T}(x)$. But even with the dynamic program for computing $p(x)$ from Proposition 4.6, the computational cost grows exponentially with the block size. As stated in Remark 4.5, we cannot avoid this exponential cost without bias. To nevertheless provide a scalable alternative to $\mathcal{L}_{\text{ratio}}$ and $\mathcal{L}_p$, we use the $k$-sample Monte-Carlo estimator

$$\log \widetilde{p}(x) := \log \frac{1}{k} \sum_{i=1, \pi_i \sim \Pi}^{k} p(x; \pi_i). \qquad (23)$$

For $k \to \infty$, clearly $\log \widetilde{p} = \log p$, so the choice of $k$ allows us to control the admitted bias of our estimator. We can then define a scalable version of $\mathcal{L}_p$ as

$$\mathcal{L}_k = \mathbb{E}_p \left[ \left( \log \frac{\widetilde{p}(x)}{\widetilde{q}_T(x)^T} - \log \frac{\widetilde{p}(x')}{\widetilde{q}_T(x')^T} \right)^2 \right], \qquad (24)$$

where for $k \to \infty$, we have $\mathcal{L}_k = \mathcal{L}_p$. In practice, it might be useful to choose different values $k$ for $\log \widetilde{p}$ and $\log \widetilde{q}_T$, as outputs of $p$ can be precomputed, allowing larger batches.

## 6. Experiments

In this section, we demonstrate the effectiveness of our approach in language modeling and aim to answer the following questions empirically:

- Which *actual* target temperatures $T$ can we achieve with our objectives and myopic TS, respectively, and how much does this degrade model performance?

- Are our variance-based objectives influenced by *likelihood drift* without the default SUMM objective $\mathcal{L}_{\text{diff}}$?

- For block diffusion, how does our procedure influence the schedule variance? How does the exact computation of $p(x)$ compare in performance to estimated likelihoods with $\mathcal{L}_k$?

- How is downstream task performance, like reasoning, influenced by our fine-tuning procedure?

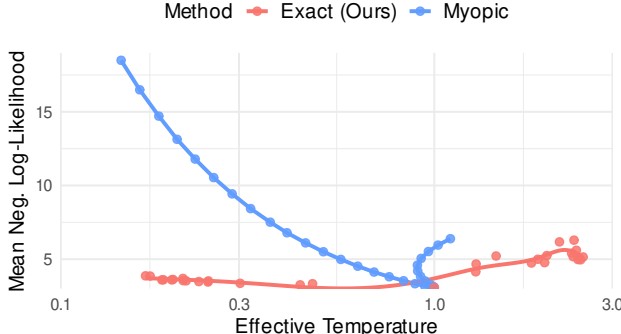

*Figure 2.* Comparison between our proposed method and tokenwise (myopic) temperature scaling with GPT2 measured on the Openwebtext dataset. Models were biased towards the temperatures 0.1 and 3.0. We show mean NLL (lower is better). Fine-tuning is able to scale to low *and* high temperatures with reduced performance loss.

**Setup** To address these questions, we compare our fine-tuned models against myopic temperature-scaling for block diffusion and autoregressive models. To compare our metrics, we fine-tune the pretrained BD3LM from Arriola et al. (2025) and GPT2 (Radford et al., 2019) on the OpenWebText corpus (Gokaslan & Cohen, 2019) using either $\mathcal{L}_p, \mathcal{L}_k$ or just myopic scaling. We target extreme temperatures $T = 0.1$ and $T = 3$ to investigate the achievable range and to assess the calibration of our procedure and myopic scaling at a temperature of $T = 0.8$. All models are trained on a dataset of 30000 sequences on a A100-80G GPUs.

In a second experiment, we investigate how well this procedure affects downstream task performance. For this, we use the two autoregressive language models Qwen-2.5-7B (Qwen et al., 2025) and the Phi-3.5-mini-instruct (Abdin et al., 2024). We fine-tune both models on the MATH dataset (Hendrycks et al., 2021) and then evaluate their accuracy on MATH500 (Lightman et al., 2024) and on HumanEval (Chen et al., 2021).

For all our experiments, we observe $T_{\text{eff}}$ (Definition 5.2), $\tau$ (Equation (13)) as well as sequence-NLL. For the diffusion experiments, we also observe the variance of $\log p(x, \pi)$ over different schedules.

### 6.1. Results

**Effective Temperature Range** We investigate how much our approach can change model behavior with extreme temperature targets. The results of this task are visualized in Figures 1a and 2 and Table 4. Myopic TS (baseline) effectively sharpens the distributions, reaching effective temperatures close to 0.1, but interestingly, does not achieve a *flatter* distribution with $T_{\text{eff}} > 1$. Furthermore, for diffusion, low temperatures lead to a drastic increase in schedule variance. In comparison, both $\mathcal{L}_k$ and $\mathcal{L}_p$ achieve better effective tem-

*Table 3.* Results on MATH500 and HumanEval. Accuracy is averaged over 5 seeds for each dataset. The shown values are pass@1 accuracies, averaged over 5 seeds. We include a run with $T = 1$ as an ablation study to test if the sequence-level objective improves model performance without actual TS.

| Model | Dataset | Base | Myopic | $\mathcal{L}_p, T = 1$ (ours) | $\mathcal{L}_p, T = 0.25$ (ours) |
|---|---|---|---|---|---|
| Phi-3.5-mini-instruct | MATH500 | 0.330 | 0.446 | 0.417 | **0.452** |
| | HumanEval | 0.066 | 0.035 | **0.101** | 0.095 |
| Qwen2.5-7B | MATH500 | 0.214 | 0.376 | **0.419** | 0.399 |
| | HumanEval | 0.296 | **0.539** | 0.318 | 0.370 |

| Model | Name | $T_{\text{eff(min–max)}}$ | $\text{NLL}_{.5}$ | $\text{Var}_{.5}$ | $\tau_{.5}$ |
|---|---|---|---|---|---|
| BD3LM | Myopic | 0.158–1.04 | 5.28 | 624.3 | **0.862** |
| | $\mathcal{L}_{k=1}$ | 0.178–2.25 | 3.41 | 134.6 | 0.851 |
| | $\mathcal{L}_{k=4}$ | 0.162–2.24 | **3.39** | 118.0 | 0.856 |
| | $\mathcal{L}_p$ | **0.143–2.36** | 4.24 | **72.3** | 0.750 |
| GPT2 | Myopic | 0.145–1.10 | 5.50 | — | **0.888** |
| | $\mathcal{L}_p$ | 0.169–2.50 | **3.33** | — | 0.803 |

*Table 4.* Results of effective temperature range. One model was fine-tuned for low/high temperature, respectively, and intermediary checkpoints were used to produce the results. The objective was combined with the SUMM objective $L_{\text{diff}}$. $T_{\text{eff}}$ shows the effective minimum and maximum temperature ratios. NLL is normalized over the sequence length, variance is averaged over log-likelihoods.

| Model | Name | $\mathcal{L}_{\text{Diff}}$ | $T_{\text{eff}}$ | NLL | Variance | $\tau$ |
|---|---|---|---|---|---|---|
| BD3LM | Myopic | | 0.93 | 3.15 | 70.15 | **0.964** |
| | $\mathcal{L}_{\text{SSTS}}$ | | 0.97 | **3.04** | 46.39 | 0.958 |
| | $\mathcal{L}_{k=1}$ | × | 0.85 | 3.23 | 69.76 | 0.948 |
| | | ✓ | 0.91 | 3.11 | 56.85 | 0.955 |
| | $\mathcal{L}_{k=4}$ | × | 0.84 | 3.25 | 81.82 | 0.948 |
| | | ✓ | 0.90 | 3.11 | 60.58 | 0.961 |
| | $\mathcal{L}_p$ | × | **0.80** | 3.44 | 86.73 | 0.945 |
| | | ✓ | 0.93 | 3.07 | **40.06** | 0.960 |
| GPT2 | Myopic | | 0.94 | 3.19 | — | **0.953** |
| | LHTS | | 0.89 | **3.10** | — | 0.809 |
| | $\mathcal{L}_p$ | × | **0.86** | 3.29 | — | 0.927 |
| | | ✓ | 0.90 | 3.11 | — | 0.936 |

*Table 5.* Results of calibration experiment losses. The models were fine-tuned with $T = 0.8$. Myopic temperature scaling is used as the baseline. NLL is normalized over the sequence length. Variance is averaged over the log-likelihoods.

perature ranges, with much lower NLL, and do not blow up the schedule variance. The choice of $k$ shows that it influences variance; however, NLL is low even for $k = 1$, indicating that this loss is highly feasible with the biased objective. Surprisingly, while $\mathcal{L}_p$ reduces schedule variance most, it also significantly reduces $\tau$. The results for GPT2 exhibit the same pattern.

**Target Temperature Precision** We investigate how precisely a specific temperature can be achieved to test the calibration of our approach. Table 5 presents the results. Across the different models, $\mathcal{L}_p$ generally changes the effective temperature most. The token-wise scaling does not seem to translate directly to changes at the sequence level, and the KL-based objectives $\mathcal{L}_{\text{SSTS}}$ and LHTS (Shih et al., 2023) only achieve minor changes in temperature. The effect of using $\mathcal{L}_{\text{diff}}$ is uniform: it prevents likelihood drift and lower NLL, but it also reduces the effect of temperature scaling. As in table Table 4, the choice of $k$ leads to improved approximation of $\mathcal{L}_p$, but seems to influence the NLL less.

**Reasoning** As a final experiment, we investigate reasoning tasks as an application of our method: We apply our fine-tuning procedure to more modern reasoning models and evaluate their downstream performance on Math500 and HumanEval. Table 3 shows the results. The effect is greatly model-dependent: Phi-3.5 benefits more from fine-tuning than myopic TS in both tasks. This is surprisingly

more pronounced in HumanEval, which is not well-aligned with the chosen fine-tuning dataset. For Qwen-2.5, this is different: Myopic TS greatly boosts performance, especially on HumanEval. In summary, the effect of either technique on downstream tasks varies widely across models and tasks.

## 7. Conclusion

In this work, we present a unifying view on TS for sequence generation models. The key ingredients for both enforcing and quantifying TS are a well-defined notion of sequence likelihood when generation schedules are stochastic and a tractable and objective metric. With these two formalisms, we define multiple objectives for temperature scaling.

Remarkably, our likelihood-ratio-based objectives $\mathcal{L}_{\text{ratio}}$ and $\mathcal{L}_{\text{schedule}}$ show much more stable performance in practice compared to vanilla importance sampling and myopic temperature scaling. Future work could investigate principled guarantees and approximations for sequence likelihood in parallel settings. Future work could also further investigate the effect of TS on downstream tasks such as molecular generation.

## Acknowledgements

PB acknowledges the support by the Austrian Science Fund (FWF) 10.55776/DOC1345324. VG acknowledges the support from the Research Council of Finland (grant 342077), Saab-WASP (grant 411025), and the Jane and Aatos Erkko Foundation (grant 7001703).

## Impact Statement

This paper presents work whose goal is to advance the field of machine learning. There are many potential societal consequences of our work, none of which we feel must be specifically highlighted here.

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

## A. Proofs

**Proposition 4.2.** *Absorbing D3PM (Austin et al., 2023), MDLM (Sahoo et al., 2024), MD4 (Shi et al., 2024), RADD (Ou et al., 2025) are SSUMs.*

*Proof.* When showing that the models are SSUMs, we need to show that they full fill the following conditions for all $\pi$ with $\Pr(\Pi = \pi) > 0$:

- $\pi$ is strictly monotone, i.e., each position is unmasked exactly once (no remasking).

- $p(x_i \mid x_S)$ depends only on $i$ and $x_S$.

- $\Pi$ is independent of the token values $x$.

We show this for MDLMs, for the others it follows equivalently. Let $\Pi$ be a random variable describing the distribution of the unmasking schedule. By definition MDLM once a token is unmasked can not be changed, therefor $\pi$ is strictly monotone for all $\pi$ with $\Pr(\Pi = \pi) > 0$. The model is only parameters by $x_S$ and the unmasking probability is independent

First, we show that $\alpha$ indeed defines a distribution for $\Pi^\alpha$. Sahoo et al. (2024) introduce the noising schedule $\alpha[0, 1] \to [0, 1]$ as a strictly decreasing function in continuous time $t$ with $\alpha_0 = 1$ and $\alpha_1 = 0$, as the generation is modeled in reverse time. During the generation, a masked token has the probability of getting unmasked in the interval $[s, t]$ of $\frac{\alpha_s - \alpha_t}{1 - \alpha_t}$ and a probability of staying masked of $\frac{1 - \alpha_s}{1 - \alpha_t}$.

Based on $\alpha$ we can define random unmasking times for each position in our sequence. We do this for discrete time where $K$ is the number of time steps, let $t_k$ for $k \in [0, K]$ be the discrete time steps. Let $\tau_i \in [0, 1]$ be the random unmasking time of the $i$th position, then it is i.i.d. distributed

$$\Pr(\tau^i = t_k) = \left( \frac{\alpha_{t_k} - \alpha_{t_{k-1}}}{1 - \alpha_{t_{k-1}}} \right) \prod_{i=1}^{k-1} \left( \frac{1 - \alpha_{t_k}}{1 - \alpha_{t_{k-1}}} \right) = \alpha_{t_k} - \alpha_{t_{k-1}}. \tag{25}$$

$$Pr(\tau^i = t_k) = f_\alpha(s, t) \prod_{i=1}^{k-1} \left( \frac{1 - \alpha_{t_k}}{1 - \alpha_{t_{k-1}}} \right) = \alpha_{t_k} - \alpha_{t_{k-1}}. \tag{26}$$

Clearly, we can map the full vector of unmasking times $\tau$ to a schedule $\pi$, that unmasks token position by decreasing time, we will call this mapping $f$. This approach is consistent, as $p(x; \tau) = p(x; \tau')$ whenever $f(\tau) = f(\tau')$ because the probabilities under the model are computed time independent, with only depending on the order. Consequently, $\alpha$ defines a valid distribution for $\Pi^\alpha := f(\tau)$. Since $\tau_i$ is i.i.d. the probability $\Pr(\pi)$ also does not depend on the positions.

Next, we will show that the probability $p(x)$ of a sequence generated by the model is the expectation over the unmasking schedules. For this we use the law of total probability, marginalizing for each time step $k \in [0, K]$ over the possible intermediate states $x_{\pi_k}$, given us

$$p(x) = \sum_{x_{\pi_1}, \dots x_{\pi_k}} \prod_{k=1}^{K} p(x_{\pi_k} \mid x_{\pi_{k-1}}) = \sum_\pi \Pr(\pi) p(x; \pi) = \mathbb{E}_\Pi[p(x; \pi)] \tag{27}$$

$\square$

**Corollary 4.4.** *Absorbing D3PM, MDLM , MD4, RADD over continuous time unmask according to $\Pi \sim \mathcal{U}(S_L)$, i.e., a sequence $x \in \mathcal{X}^L$ is unmasked sequentially and*

$$\Pr(x) = \mathbb{E}_{\Pi \sim \mathcal{U}(S_L)}[p(x; \Pi)] = \frac{1}{L!} \sum_\pi p(x; \pi). \tag{10}$$

*Proof.* For continuous time, the probability of unmasking two tokens at the same time is zero, as it is equivalent to sampling discrete points (time steps) $t_1, t_2 \sim \mathcal{U}[0, 1]$ which is a null set. This means that for all $\pi$ only one token get unmasked at a time, so the only thing distinguishing the $\pi$ is in which order the tokens are getting unmasked. As $\Pr(\pi)$ is independent of time and position, it is the same for all $\pi$. Giving us a uniform distribution over the symmetric group $\Pi \sim \mathcal{U}(S_L)$. $\square$

**Proposition 4.6.** *Let $p$ be a block diffusion model, with block size $B$, in a sequential unmasking regimen. Then, for a given $x$, we can compute $p(x)$ in time $O(L2^B)$.*

*Proof.* We assume we can choose $M \in \mathbb{N}$ such that $L = MB$. For each token position $i \in [L]$, there are $2^{B-1}$ possible partial sequences $x_S$ when $i$ is unmasked. This means $p(x)$ can be written in terms of exactly $L2^{B-1}$ unique factors $p(x_i \mid x_S)$, each of which requires one forward pass through the model $p$. Then $p(x)$ can be computed using the dynamic programming method illustrated in Algorithm 1.

$\square$

**Proposition 5.3.** *Let $p$ be a pre-trained MDM, sampled according to $\Pi$, and $\mathcal{D} \subset \mathcal{X}^L$ be a data set of sequences. If $\forall x \in \mathcal{D}$ and $\forall \pi_i$ with $\mathrm{Pr}_\Pi(\pi_i) > 0$ it holds that $\mu(x_{\pi_i})$ is finite, then $\mathcal{L}_{SSTS}$ minimizes the KL-Divergence $KL(p_T \| q_T)$ and is a strictly proper loss, it is minimal only if $q_T \propto p^{1/T}$.*

*Proof.* The $\mathcal{L}_{\text{diff}}$ minimizes the KL Divergence $D_{KL}(p\|q)$ by factorizing over prefixes and minimizing the conditional KL Divergence (Shi et al., 2024). As Shih et al. (2023) shows adding the importance weight minimizes the KL Divergence $D_{KL}(p_T \| q_T)$ to the temperature scaled distribution $p_T = p^{1/T}$, therefore our loss $\mathcal{L}_{SSTS}$ minimizes the KL Divergence.

To show that our loss is strictly proper we rewrite it:

$$\mathcal{L}_{\text{LHTS}} = -\mathbb{E}_{\pi \sim \mathcal{U}(S_L)} \mathbb{E}_{i, x_{\pi_i} \sim p} \exp(-\mu(x_{\pi_i})) \left[ \sum_{i=1}^{L} \exp\left( \frac{1-T}{T} \log p(x \mid x_{\pi_{i-1}}) \right) \log q_T(x_{\Delta_i} \mid x_{\pi_{i-1}}) \right]. \quad (28)$$

The inner expectation defines a strictly proper loss as shown by Shih et al. (2023). Since a positive combination of strictly proper losses is also a proper loss, $\mathcal{L}_{\text{LHTS}}$ is also a strictly proper loss. $\square$

**Proposition 5.4.** *Let $p$ be a pretrained SUMM with global support over $\mathcal{X}^L$, and let $T \in \mathbb{R}_+$ be the target temperature. A model $q_T$ that minimizes $\mathcal{L}_{ratio}$ minimizes the* relative entropy variance *$V(p_T \| q_T)$ (Hayashi, 2009). When $V(p_T \| q_T) = 0$, it holds that $KL(p_T \| q_T) = 0$.*

*Proof.* We define $h(x) = \log p(x) - T \log q_T(x)$. Then it holds that

$$\log \frac{p(x)}{p(x')} - T \log \frac{q_T(x)}{q_T(x')} = \log p(x) - \log p(x') - T \log q_T(x) + T \log q_T(x') \quad (29)$$

$$= \log p(x) - T \log q_T(x) - (\log p(x') - T \log q_T(x')) \quad (30)$$

$$= h(x) - h(x') \quad (31)$$

$$\quad (32)$$

Consequently, $\mathcal{L}_{\text{ratio}} = \mathbb{E}_{x, x' \sim p_T}[h(x) - h(x')]^2 = \mathrm{Var}_{p_T} h(x)$. Importantly, $\mathrm{Var}_{p_T} h(x) = V(p\|q^T) + c'$, where $V$ is the relative entropy variance (Hayashi, 2009) of our two distributions, so a model $q_T$ that minimizes $\mathcal{L}_{\text{ratio}}$ minimizes the $V(p_T \| q_T)$.

If $\mathcal{L}_{\text{ratio}} = 2\mathrm{Var}_{p_T} h(x) = 0$, we have that $h(x) = c$ for some constant $c \in \mathbb{R}$. That is, it holds

$$\log p(x) = T \log q_T(x) + c \quad (33)$$

$$p(x)^{1/T} = \exp(c) q_T(x) \quad (34)$$

This is the definition of temperature scaling, enforced over the distribution $D$. For normalized $p$ and $q$, and when the support of $p$ is global, we have $\exp(c) = Z_{p_T}$. The normalized $p_T$ is then $p_T = \frac{p(x)^{1/T}}{\exp(c)} = q_T$. It follows that $KL(p_T \| q_T) = 0$.

$\square$

# B. Experiments

Code is available at `github.com/Aalto-QuML/temp_lm` with all details for reproducibility.

