# OpenReview forum: "Temperature Scaling in Discrete Sequence (Language) Models"
_ICML.cc/2026/Conference — ICML 2026 regular_

### Official Review · Reviewer_g7jK · 2026-03-17

**Soundness:** 3
**Presentation:** 2
**Significance:** 3
**Originality:** 3
**Overall Recommendation:** 3
**Confidence:** 3

**Summary:**

This paper assesses a general area of sequence-level temperature scaling for discrete sequence models, especially masked/diffusion-style models where the probability of a sequence depends on the random unmasking order rather than a single canonical factorization. The paper’s main claim is that naive token-wise temperature scaling is fundamentally wrong at the sequence level, and that this problem is particularly severe for masked diffusion models because likelihood estimates vary substantially across unmasking schedules. To address this, the authors introduce a unified formalism of stochastic sequence unmasking models (SSUMs), give an exact dynamic program for computing sequence likelihoods in block diffusion settings, propose a partition-free temperature-scaling metric based on likelihood ratios, and develop several fine-tuning objectives for stochastic sequence temperature scaling, including an importance-sampling objective and two lower-variance ratio/schedule regularizers.

**Compliance With Llm Reviewing Policy:**

Affirmed.

**Key Questions For Authors:**

Can you better disentangle whether the gains come from sequence-level temperature matching itself versus simply adding a stabilizing fine-tuning regularizer on top of the base diffusion loss?

Do the effective-temperature improvements translate into better generation behavior on any downstream task beyond lower NLL/variance, for example diversity-quality tradeoffs judged by external evaluators or task metrics? Also, since Table 3 and Table 4 suggest somewhat different tradeoffs for L ratio, L schedule, and LSSTS, could you clarify which objective you view as the main practical recommendation and under what regime is one preferred over the others?

**Limitations:**

Yes

**Strengths And Weaknesses:**

Strengths:
The problem of temperature scaling is ubiquitous in practice, but for sequence models with stochastic generation order, the standard token-wise view is not theoretically justified, so formalizing what temperature even means is already a worthwhile contribution. I think the strongest aspect is the conceptual package: the SSUM formalism is clean, the observation that exact sequence likelihood should marginalize over schedules is important, and the partition-free likelihood-ratio criterion is a clever way to evaluate scaling without needing the intractable partition function. The authors proceed to investigate an important concept with both theoretical and practical components, rather than offering only an empirical heuristic. I also like that the paper does not stop at the importance-sampling extension of LHTS, but proposes ratio-based regularizers precisely because the straightforward objective is unstable. Empirically, the results do support the qualitative claim that fine-tuned sequence-level scaling is more stable than myopic temperature scaling, particularly in variance and in the achievable high-temperature regime.

Weaknesses:
My main concern is that the paper’s empirical validation is still fairly limited relative to the breadth of its claims. Almost everything is demonstrated on BD3LM with one sequence length/block configuration, so it is hard to know whether the framework generalizes beyond this fairly specific setting. The theory also has some scope restrictions that matter in practice, like the exact likelihood computation is developed for sequential unmasking in block diffusion, while the motivating discussion is broader and includes more general masked diffusion settings, so the gap between the general formalism and the tractable training/evaluation regime should be clarified. I was also not fully convinced by the importance-sampling objective, since the paper’s own results suggest it barely changes the effective temperature while retaining the instability one would worry about. This makes the practical contribution rest mostly on the regularized objectives rather than on the “principled” KL-minimizing one.

---

> ### Author Rebuttal · Authors · 2026-03-31
>
> We thank the reviewer for their interest in our work and are happy our unifying formalism for sequence models in Section 4 resonates with them.
> The summary of our conceptual strengths is very sharp and we will use this phrasing to improve the presentation of our results.
>
> Indeed, we hope to address the concerns raised by the reviewer by clarifying the different properties of the loss functions.
> In addition, we report additional experimental results for autoregressive models (GPT2) here.
>
> # Trade-off and Motivation of the different loss functions
>
> You pointed out that the SSTS loss does not effectively scale the temperature, this is indeed expected.
> We summarize the three presented objective functions from the paper with a clear statement about their properties and trade-offs:
>
> - $L_{SSTS}$ is the principled "true" target for our objective, a KL minimizer, using the DP. However, training is unstable and we cannot optimize well for this directly. A detailed explanation can be seen in our response to KaMP.
> - $L_{ratio}(+\beta L_{diff})$ is the partition-free approach to enforce the temperature scaled behavior, still with the DP. $L_{ratio}$ does not suffer from any of the stability pathologies for $L_{SSTS}$, but is in itself not a proper loss. We use the vanilla objective with small beta to "anchor" it. Because of the use of the DP, the runtime is bottlenecked for large $B$.
> - $L_{schedule}(+\beta L_{diff})$ alleviates the dependence on the DP during training (the likelihoods for the base model can be pre-computed). It inherits the advantageous stability of $L_{ratio}$ but does not enforce the variance-reducing behavior, leaving inherent schedule-variance of block diffusion models as a source for performance loss during temperature scaling.
>
> # New Experimental Results with GPT2 show same observations
>
> We are happy to report new experimental in an autoregressive setting with GPT-2.
>
> In response to the reviewer’s request , we repeated the experiments of Section 6 in an autoregressive setting replacing the model with GPT2 while keeping dataset, training and evaluation setup unchanged. For comparison we fine-tune the same GPT2 model with the same dataset and compute budget using the LHTS loss of [1]. The results are provided **[here](https://drive.google.com/file/d/1x3tyzFh2csI98SW-AAHc_dLIIILWECQv/view?usp=drive_link)**.
>
> To evaluate the effective temperature range, we compare our learning objective $L_{ratio} + 0.05L_{AR}$, against the myopic baseline at temperatures 0.1 and 3.0.
>
> The results in Table 1 show that our method is more effective in temperature scaling than the LHTS model and Figure 1 shows that it gives a better temperature performance trade-off compared to the myopic temperature scaling. These results suggest that the method extends beyond the block-diffusion setting to autoregressive models.
>
> In addition we extended the experimental results and evaluated our method on the reasoning benchmark MATH500, showing improved accuracy in downstream tasks, for further details we refer to the response to j8Xm and the [results](https://drive.google.com/file/d/1x3tyzFh2csI98SW-AAHc_dLIIILWECQv/view?usp=drive_link).
>
> # Questions
>
> ## Q1: Sequence Level Temperature Matching vs. Stabilizing Regularizer: Improved Performance even at T=1
>
> To answer this question, we included ablation results for the new results on reasoning, and will do full ablations for the camera ready version.
> Our training regime was to train the model once with $L_{AR}$, once with $L_{AR}$ + $L_{ratio}$ for $T=1$ and $T=0.25$ respectively.
>
> Interestingly, even without explicitly changing the temperature, the regularizer improves the performance of the base model in MATH500 slightly from 0.395 to 0.412.
> However, the effect of choosing T=.25 then shows a much more drastic improvement to 0.460.
>
> So using $L_{ratio}$ only as regularizer shows some beneficial effect for downstream tasks, but explicit temperature scaling shows much more drastic improvement.
>
>
> ## Q2: Tradeoffs between objectives and practical recommendation.
>
> We hope that our clarification above already partially answered this question: While theoretically the most sound, importance sampling with $L_{SSTS}$ is often unstable in practice.
> $L_{ratio}$ is the stable partition-free proxy to this.
> Both these objectives stabilise the high variance of the model, requiring likelihood computation.
> $L_{schedule}$ then relieves the cost incurred by the likelihood computation, as the DP is only needed for the constant base model and can be precomputed.
>
> In practice, we recommend either the use of $L_{ratio}$ or $L_{schedule}$, depending on the desired tradeoff between training-cost and schedule-variance reduction.
>
> Acting on your feedback has helped reinforce the strengths of this work. We hope the same is reflected in your upgraded score. We thank you again for the feedback. Should there be any other concerns or questions, we are happy to engage further.

---

> > ### Author Rebuttal · Reviewer_g7jK · 2026-04-05
> >
> > The authors answered my questions, I adjust my score to a 4.

---

> > > ### Author Response · Authors · 2026-04-05
> > >
> > > We thank the reviewer for their response and are happy that we have resolved their concerns.
> > > We are especially happy that our additional experimental results are convincing and demonstrate that our experimental interpretation holds across different model classes and meaningfully impacts quality in downstream tasks.
> > >
> > > Should there be any further concerns that hold the manuscript back, we are happy to address them and engage in further discussion.
> > > Otherwise, we would respectfully ask the reviewer to reflect their response by updating the original score. Thank you!

---

### Official Review · Reviewer_j8Xm · 2026-03-20

**Soundness:** 2
**Presentation:** 2
**Significance:** 2
**Originality:** 2
**Overall Recommendation:** 2
**Confidence:** 4

**Summary:**

This paper extends long-horizon temperature scaling to text diffusion models, but mostly block-diffusion models where one can compute exact likelihoods of blocks. Since the presence of blocks prevents applying the original variance reduction scheme of LHTS, the paper proposes a surrogate loss similar to ratio matching and a schedule-conditioned version.

Experiments show that the surrogate losses outperform the original importance-weighted objective when attempting to recover the ground truth temperature of block diffusion models. However, the true temperature is not actually recovered. When training to calibrate to a range of temperatures, $L_ratio$ appears to be the most stable across a range of temperatures. a

**Compliance With Llm Reviewing Policy:**

Affirmed.

**Key Questions For Authors:**

The paper has positive preliminary results, but should the experiments should be scaled up. Compare the experiments to the original LTHS paper, which had image diffusion and language modeling at a reasonable scale. The modern incarnation of this should definitely test on reasoning, similar to: https://arxiv.org/abs/2510.14901.

**Limitations:**

Yes

**Strengths And Weaknesses:**

The paper appears to be theoretically sound, but the experimental evidence is preliminary. Results are presented only on sequences of length 128, with block size 4. More experiments on more realistic settings, such as with simple math reasoning, would be more convincing.

The presentation is fine, although in my opinion section 4 is quite trivial and does not scale beyond block diffusion with very small block size. The original LHTS paper also had experiments on image diffusion models.

In terms of significance, the empirical results with the ratio matching loss could be interesting if proven to work at more realistic scale. The application of ratio-matching losses to temperature tuning is original.

---

> ### Author Rebuttal · Authors · 2026-03-31
>
> We thank the reviewer for their positive stance toward the significance of our work (if proven on more realistic scales).
> We found the suggestion of reasoning tasks and the suggested work to be exceptionally useful, and we thank the reviewer for the suggestion.
> Our response is structured as follows: we first present new experimental results on GPT-2 and Phi-3 with MATH500, and briefly discuss the comment on our theory in Section 4.
>
> # Theoretical Results for Stochastic Sequence Unmasking Models
>
> We would like to motivate and justify our contributions a bit more.
> First, the general definition of SUMMs and their behavior is simply required to present unified theoretical results, as existing definitions are scattered and incompatible.
> We use this formalism to allow for a transfer between different sequence models without barriers.
>
> Second, we agree that the scalability of Algorithm 1 is limited, even if the runtime is drastically better than naive enumeration.
> However, it turns out that there are no straightforward "sophisticated" results (i.e. concentration bounds from polynomial samples) we can obtain for this regime.
> We still believe it is important to address this (unwanted) characteristic of block diffusion models, as it is an important source of inconsistency for temperature-scaling.
> If the reviewer has any suggestions for the improvement of our results, we would be keenly interested to hear them.
>
> # New Results on Autoregressive Models: GPT-2 shows that the observations hold at scale
>
> Based on the feedback, we applied our method to GPT-2, similar to [1], and report the full results [here](https://drive.google.com/file/d/1x3tyzFh2csI98SW-AAHc_dLIIILWECQv/view?usp=drive_link), with a summary in our response to reviewer g7jK.
> We are happy to report that the observations on BD3LM transfer nicely to the autoregressive setting, with a sustained low runtime and strong scaling performance. Our method achieves an effective temperature closer to the target compared to the LHTS method of [1].
>
> We are happy to widen the scope of our results further, but want to emphasize the focus of our work on sequence models.
> For this reason, we consider other model classes such as continuous diffusion models as a very interesting direction for future work, but try to keep our experimental evaluation focused on the unique aspects of SUMMs.
>
> # Reasoning Benchmark on MATH500 with Phi-3 shows strong improvements in downstream reasoning tasks
>
> The suggestion of a reasoning benchmark, and especially a paper that has the same objective as us, is extremely helpful, and we want to thank the reviewer for this note.
>
> We ran a part of the experiments from [2] and report them [here](https://drive.google.com/file/d/1x3tyzFh2csI98SW-AAHc_dLIIILWECQv/view?usp=drive_link). We fine-tuned Phi-3.5-mini-instruct [3] on the grade school math dataset GSM8K using our learning objective $L_\text{ratio} + 0.01 L_\text{AR}$ with temperature 0.25. For comparison, we fine-tuned two baselines, one trained with the same losses but with temperature 1.0 and one with just the standard autoregressive loss. We evaluate on the MATH500 dataset consisting of problems from mathematics competitions.
>
> |  | Baseline | Baseline (myopic temp. scaling) | Our Method |
> | ---- | --- | --- | --- |
> |  MATH500  |       0.395      |          0.444         |          **0.460**         |
>
> Table 3 shows that our method achieves the highest accuracy (0.46), compared to the temperature 1.0 baseline with accuracy 0.41 and the autoregressive baseline with accuracy 0.4.
>
> We are excited to expand on these results for the camera-ready version of our manuscript and cite the work.
> The low cost of our fine-tuning procedure compared to methods that rely on sequence rollouts like the reinforcement learning method GRPO [1] or the power sampling methods of [2] makes a compelling argument for our approach.
> Further, the capability to improve performance in a reasoning task by fine-tuning on an unrelated dataset suggests strong transferability of this approach.
> We again thank the reviewer for bringing this to our attention!
>
> Acting on your feedback has helped reinforce the many strengths of this work and ground our metrics to downstream performance.
> We hope the same is reflected in your upgraded score.
> We thank you again for the suggestion.
> Should there be any other concerns or questions, we are happy to engage further.
>
> References:
>
> [1] Shih, Andy, Dorsa Sadigh, and Stefano Ermon. "Long horizon temperature scaling." International conference on machine learning. PMLR, 2023.
>
> [2] Karan, Aayush, and Yilun Du. "Reasoning with sampling: Your base model is smarter than you think." arXiv preprint arXiv:2510.14901 (2025).
>
> [3] Abdin, Marah et al. “Phi-3 Technical Report: A Highly Capable Language Model Locally on Your Phone.” ArXiv abs/2404.14219 (2024).
>
> [5] Shao, Zhihong, et al. "Deepseekmath: Pushing the limits of mathematical reasoning in open language models." arXiv preprint arXiv:2402.03300 (2024).

---

### Official Review · Reviewer_KaMP · 2026-03-24

**Soundness:** 2
**Presentation:** 2
**Significance:** 2
**Originality:** 2
**Overall Recommendation:** 3
**Confidence:** 3

**Summary:**

This paper addresses token-wise temperature scaling, known to distort the MAP estimate at the sequence level, but applies principled sequence-level temperature scaling to masked diffusion models (MDMs). The paper makes three interconnected contributions: (1) a unified formalism, Stochastic Sequence Unmasking Models (SSUMs), that subsumes autoregressive models, MDMs, and block diffusion under a common likelihood definition; (2) a partition-free metric for measuring effective temperature scaling; and (3) three fine-tuning objectives for SSTS. Experiments are conducted on BD3LM with L=128, B=4 on OpenWebText.

**Compliance With Llm Reviewing Policy:**

Affirmed.

**Ethical Review Concerns:**

N/A.

**Final Justification:**

The rebuttal addressed some of my concerns, and I have raised my score from 2 to 3. However, I still feel the paper lacks sufficient novelty for ICML. The theoretical contributions are mostly notational, the principled objective (L_SSTS) doesn't work in practice, and the methods that do work are relatively straightforward variance regularizers. All these convinced me for a weak reject score.

**Key Questions For Authors:**

I have already included my detailed questions and points for the authors in the weaknesses section. Please refer to that section.

**Limitations:**

Yes.

**Strengths And Weaknesses:**

This paper addresses a theoretical gap and makes the following contributions: The SSUM unification is useful, the partition-free metric is principled, and the connection between variance regularization and ratio matching is compelling. However, in its current form, the paper overstates its practical achievements, which I detail as follows:

1. The importance-sampling objective LSSTS is the paper's direct analog of LHTS to SSUMs. Table 3 shows that LSSTS achieves an effective temperature of exactly **1.00** (unchanged from the base model) while also incurring the highest NLL. The method designed to move the model to T=0.8 does not move it at all. The paper notes this result but provides no satisfying mechanistic explanation.

2. All experiments use a single model (BD3LM) with a single block size (B=4) and a single sequence length (L=128). This is far too narrow given the paper's stated contributions. The authors could have included experiments on full MDM or autoregressive models.

3. Section 6 is a bit misleading. Although the fine-tuned models cover a wider range, at low temperatures the baseline is clearly better, reaching 0.226 vs 0.431 and 0.556. This is exactly the regime where quality matters most. The gains are mainly at high temperatures, 2.85 vs 1.51, which is less critical. Also, Lschedule has the best NLL at T equals 0.5, but its variance is extremely high, 585 vs 31.8, making outputs unstable. Overall, the method mainly improves diversity, while the baseline remains stronger for quality.

4. Corollaries 4.4 and 4.5 are near-duplicates. This appears to be a copy-paste error from an earlier draft.

5. Algorithm 1 requires a forward pass through model p for each partial sequence state, meaning the total number of neural network evaluations is O((L/B) · 2^B). For B=4 this is 32 evaluations per sequence. For B=16 it is 65,536. The paper does not discuss the computational cost relative to training, or how this scales relative to the training budget of 30,000 sequences. The "Runtime: 34 + 34 minutes" in Table 4 does not break down these costs or relate them to training without the exact DP.

6. Proof of Proposition 5.2 contains a notation inconsistency. The proof claims "our loss LKL minimizes the KL Divergence" but LKL is never defined in the paper!!!

---

> ### Author Rebuttal · Authors · 2026-03-31
>
> We thank the reviewer for the detailed feedback and their positive words for our theoretical contributions.
>
> We respond by topic:  **behaviour of $L_{SSTS}$** (1.), **interpretation of experiments** (3.), **new experiments on GPT2 and Reasoning** (2.), and **block diffusion scalability** (5.).
> Finally, (4.,6.) are indeed typesetting artefacts that we now fixed. We have corrected $L_{KL}$ to $L_\text{SSTS}$.
> # $L_{SSTS}$: expected instability (1.)
>
> We agree $L_{SSTS}$, i.e., importance sampling **does not perform well in practice**. This is not surprising and the main reason we introduce other losses.
> The **mechanistic explanation** for this is the highly peaked weight distribution during training:
>
> For $T' = \frac{1-T}{T}$ the importance weights are defined as $w_T(x) = p(x)^{T'}<1$.
> Now, for low temperatures it holds $T'>1$ and the ratio between importance weights is sharpened, i.e., few weights remain large and the majority of weights is negligibly small.
> This causes numerical stability issues that need to be compensated with weight clipping or very small learning rate, as already mentioned in [1].
> *This limitation also applies to the work of [1]*, and motivates us to introduce surrogates for this objective to avoid all these instabilities.
> # Experimental Results in Table 4: Baseline is consistently worst (3.)
>
> We believe Table 4 in the manuscript may have been misinterpreted.
> In temperature scaling, the goal is not purely to reach a low temperature, but to maintain performance at the same time.
> Consequently, we care most about the **Pareto front** instead of one particular value in temperature or NLL.
>
> > At low temperatures the myopic baseline is "clearly better" (reached a lower minimal temperature).
>
> This is correct, but in Table 4, NLL.5 indicates much stronger performance loss in the baseline compared to our two methods.
> This is confirmed in Figure 1a, where the Pareto front of all method reveals that the **baseline performs worst**.
>
> > $L_{schedule}$ has the best NLL.5 but its variance is extremely high, making outputs unstable.
>
> BD3LM exhibits strong inherent variance, which $L_{schedule}$ does not improve, but does not drastically worsen, as can be seen in comparison with the baseline.
> In contrast, $L_{ratio}$ **reduces variance by more than an order of magnitude** over the baseline and $L_{schedule}$.
>
> > the method mainly improves diversity, while the baseline remains stronger for quality.
>
> The baseline (myopic temperature) scaling is **not advantageous over either of our methods**.
> Myopic temperature scaling can reach low temperatures but collapses model quality. This is the main motivator of this work.
>
> # Further experimental results on GPT2 and Reasoning Tasks for Phi-3 show strong results (2.)
>
> Based on the feedback, we now present additional results using the autoregressive GPT2, to anchor our work to [1] as well as a benchmark on math reasoning tasks, to showcase a tangible impact of our approach and metrics in downstream methods.
>
> Due to spatial constraints, we refer to the **[result summary](https://drive.google.com/file/d/1x3tyzFh2csI98SW-AAHc_dLIIILWECQv/view?usp=drive_link)** for the numeric results and to the response to reviewer j8Xm for more details, but provide a brief summary:
>
> **Our observations on BD3LM translate to autoregressive models (GPT2)**.
> Table 2 shows that we can achieve a very wide effective temperature range with GPT2, reaching temperatures as low as $0.358$ with only marginal performance cost.
> At the same time, myopic scaling to similar scales leads to a complete performance degradation.
> # Block Diffusion Scalability (5.)
>
> The complexity of Algorithm 1 for BD3LM allows for $O(2^B)$ *forward passes/space*  resulting in a *runtime* of $O(L 2^B)$, so we require 16 forward passes in our experiments.
> While this is exponential in $B$, the general motivation of block diffusion is to keep small, near-constant values for $B$.
> For settings where B is larger, the exact likelihood computation is indeed costly, which is one motivation for our $L_{schedule}$, which is independent.
>
> The overhead of the DP is minimal, with the cost being dominated by the forward passes of the model, which we compensated with smaller batch size in training.
> Further, the DP of the base model can be pre-computed and cached, which is why $L_{schedule}$ does **not need the DP during training at all**.
> We implemented this improvement and reran the experiments, showing that without DP, the runtime is 14 minutes for $L_{schedule}$ versus 34 minutes for $L_{ratio}$, where we use the DP during training. These results are in the attached table 5.
>
> If our broader experimental evaluation and the clarifications of our motivation and results addressed all mentioned weaknesses, we would be more than grateful for a reflection of this in the evaluation of the reviewer.
>
> [1] Shih, Andy, Dorsa Sadigh, and Stefano Ermon. "Long horizon temperature scaling." International conference on machine learning. PMLR, 2023.

---

> > ### Author Rebuttal · Reviewer_KaMP · 2026-04-07
> >
> > The new GPT-2 and MATH500 experiments partially address my main concern about narrow evaluation, and the typo fixes are noted. However, the MATH500 results cover only one temperature/loss combination; the high variance of L_schedule is not convincingly contextualized against the base model's inherent variance.
> > Given the added experiments and clarifications, I am raising my score to 3.

---

> > > ### Author Response · Authors · 2026-04-08
> > >
> > > We thank the reviewer for their additional feedback. We address your further concerns below.
> > >
> > > # Further evidence: New experimental results
> > > Thank you for your constructive suggestion of varying the temperature for our reasoning experiments. We repeated the reasoning experiment with different target temperatures. As for autoregressive models there is only one unmasking schedule $\pi^{AR}$ $L_{ratio}$ and $L_{schedule}$ are the same. So we only have one loss for the experiments. Our results are provided in the table below.
> > >
> > >
> > > | Model                                         |   MATH500 |
> > > | --------------------------------------------- | --------: |
> > > | Baseline (transfer)                           |     0.395 |
> > > | myopic temperature scaled Baseline (transfer) |     0.444 |
> > > | Sequence Temp. Scaling temp = 1.0 (ours)      |     0.412 |
> > > | Sequence Temp. Scaling, temp = 0.75 (ours)    |     0.428 |
> > > | Sequence Temp. Scaling, temp = 0.5 (ours)     | **0.468** |
> > > | Sequence Temp. Scaling, temp = 0.3 (ours)     |     0.444 |
> > > | Sequence Temp. Scaling, temp = 0.25 (ours)    |     0.460 |
> > > | Sequence Temp. Scaling, temp = 0.2 (ours)     |     0.454 |
> > > | Sequence Temp. Scaling, temp = 0.1 (ours)     |     0.436 |
> > >
> > > Thus, repeating the experiment with different target temperatures led to an improvement of the MATH500 accuracy, further widening the gap between the myopic baseline and our approach. The model with target temperature 0.5 reaches the best accuracy of 0.468. Furthermore the results suggest that our method preforms well over different target temperatures.
> > >
> > > Even though the benchmark is slightly noisy, the results are consistent with the observations by  Karan and Du (2025), suggesting that the improvements to reasoning capabilities seem maximal at some non-zero temperature and then slightly decrease.
> > >
> > > We emphasise that in contrast to the method by  Karan and Du (2025), our method only needs one (low cost) fine-tuning run and then has no extra sampling costs. This makes our method a more compute efficient alternative to archive a temperature scaled distribution and the resulting reasoning improvements. We will include a discussion on this in the camera-ready.
> > >
> > > # Contextualization of the Variance of $L_{schedule}$ with respect to the base model
> > >
> > > We expect the variance of a model to increase strongly when we decrease its temperature.
> > > To show this, we can make the following rough observations.
> > >
> > > For a given model $p$, the $\operatorname{Var}$ in the table reports the quantity
> > > $\\operatorname{Var}(p) \\coloneqq \frac{1}{n}\\sum_{i=1}^{n} \\operatorname{Var}_{\pi}\\left[\\log p(x;\\pi)\\right]$ .
> > >
> > >
> > > Let $q = p^{1/t}$ denote the temperature-scaled model for constant $0<t<1$. Then we have $\frac{1}{n}\sum\limits\_{i=1}^{n}\textrm{Var}\_{\pi}\log q(x;\pi) = \frac{1}{n}\sum\limits\_{i=1}^{n}\textrm{Var}\_{\pi}\frac 1t\log p(x;\pi)= \frac{1}{n}\sum\limits_{i=1}^{n}\frac 1{t^2}{\textrm{Var}\_{\pi}\log p(x;\pi)}= \frac{1}{t^2}\frac{1}{n}\sum\limits\_{i=1}^{n}\textrm{Var}\_{\pi}\log p(x;\pi)$
> > >
> > > This means that $\mathrm{Var} (q) =\frac{1}{t^2} \mathrm{Var} (p)$.
> > >
> > >
> > > As the effective temperature of the model $T$ is non-constant in practice (after training, $T$ may vary across inputs in practice), we can make a conditional statement with the same derivation instead:
> > >
> > > $\operatorname{Var}\_{\pi}\bigl(\log q(x;\pi)\mid T\bigr)=\frac{1}{T^2}\operatorname{Var}\_{\pi}\bigl(\log p(x;\pi)\mid T\bigr)$
> > > and assuming the variance of $p$ is roughly independent of $T$, we can consequently say $$ \operatorname{Var}(q)\approx \mathbb{E}\left[\frac{1}{T^{2}}\right]\operatorname{Var}(p)\geq
> > > \frac{\operatorname{Var}(p)}{\mathbb{E}[T]^{2}} \approx \frac{\operatorname{Var}(p)}{t^{2}} $$
> > > While this derivation makes simplifying assumptions, it showcases that high variance at low temperatures is expected, and consequently exhibited by both the myopic baseline and $L_{schedule}$. The $L_{schedule}$ has similar variance to the myopic baseline as Figure 2 and Table 4 and 5 show. The results in Figure 2, and Table 4 and 5 suggest that the $L_{ratio}$ does not only effectively temperature scale the model but also prevents this blowup in variance.
> > >
> > > We hope that you will consider raising your score in the light of the additional empirical results and variance contextualization. Many thanks!

---

### Decision · Program_Chairs · 2026-04-30

**Decision:**

Accept (regular)

**Comment:**

This paper studies the problem of temperature scaling in language models (considering autoregressive and masked diffusion LMs under a single paradigm), where naive token-wise temperature scaling is not well defined at the sequence level.

Reviewers agree that the problem being studied is interesting and the results are promising. Several reviewers said that the proposed formalism is a useful unification across autoregressive and diffusion-style models, and the partition-free likelihood ratio metric is a principled way to evaluate temperature scaling without requiring access to the partition function. The primary concerns raised in the initial reviews relate to the limited empirical evaluation and the gap between the general theoretical framework and the relatively narrow experimental setting (primarily block diffusion with small block size). The rebuttal does address these concerns by providing additional rseults on autoregressive models and a reasoning task (MATH500).

While the empirical evaluation remains somewhat limited relative to the full generality of the framework, I find that the paper’s conceptual contribution is strong and sufficiently validated to warrant acceptance. I encourage the authors to include all new results in the final draft.